# A multi-resolution systematically improvable quantum embedding scheme for large-scale surface chemistry calculations

Zigeng Huang [1] ✉, Zhen Guo [1], Changsu Cao[1], Hung Q. Pham [2], Xuelan Wen [1], George H. Booth [3] ✉, Ji Chen [4,5,6] ✉ & Dingshun Lv[1] ✉

Predictive simulation of surface chemistry is critical in fields from catalysis to electrochemistry and clean energy generation. Ab-initio quantum many-body methods should offer deep insights into these systems at the electronic level but are limited by their steep computational cost. Here, we build upon state-of-the-art correlated wavefunctions to reliably reach 'gold standard' accuracy in quantum chemistry for extended surface chemistry. Efficiently harnessing graphics processing unit acceleration along with systematically improvable multi-resolution techniques, we achieve linear computational scaling up to 392 atoms. These large-scale simulations demonstrate the importance of converging to these extended system sizes, achieving consistency between simulations with different boundary conditions for the interaction of water on a graphene surface. We provide a benchmark for this water-graphene interaction that clarifies the preference for water orientations at the graphene interface. This is extended to the adsorption of carbonaceous molecules on chemically complex surfaces, including metal oxides and metal-organic frameworks, where we consistently achieve chemical accuracy compared to experimental references. This advances the simulation of molecular adsorption on surfaces, enabling reliable and improvable first-principles modeling of such problems by ab-initio quantum many-body methods.

The chemical and physical properties of material surfaces play a central role in a wide range of scientific research areas and industrial applications[1–3]. However, predictive simulations of these systems without resorting to empiricism remain a formidable and long-standing challenge, requiring an accurate solution to the many-electron problem[4–9]. The formal solution to this problem scales exponentially in computational effort with system size, and therefore to make progress necessitates a sufficiently accurate approximation to the electron correlations while also rendering the problem computationally efficient for scaling to large systems at the bulk limit. Over the past few decades, density functional theory (DFT) has emerged as the standard bearer for this, favored for its wide applicability and low computational cost[10–12]. However, DFT is not systematically improvable due to its reliance on semi-empirical exchange-correlation functionals, which are not universal and may not provide sufficient transferability or internal validation of its accuracy across different chemical environments. In contrast, correlated methods[13], such as Coupled Cluster theories[9] or quantum Monte Carlo methods[8],

[1]ByteDance Seed, Fangheng Fashion Center, Beijing, PR China. [2]ByteDance Seed, San Jose, CA, USA. [3]Department of Physics, King's College London, London, UK. [4]School of Physics, Peking University, Beijing, PR China. [5]Interdisciplinary Institute of Light-Element Quantum Materials and Research Center for Light-Element Advanced Materials, Peking University, Beijing, PR China. [6]State Key Laboratory of Artificial Microstructure and Mesoscopic Physics, Frontiers Science Center for Nano-Optoelectronics, Peking University, Beijing, PR China. ✉e-mail: huangzigeng@bytedance.com; george.booth@kcl.ac.uk; ji.chen@pku.edu.cn; lvdingshun@bytedance.com

can achieve superior accuracy by explicitly describing electron correlation, and importantly their accuracy can be systematically improved[14–19]. Among these, the coupled-cluster with single, double, and perturbative triple excitations (CCSD(T))[20], is often regarded as the 'gold standard' of electronic structure theory, and has been shown to accurately model the relevant long-range interactions on material surfaces[21–23]. However, in the worst case, the correlation present within the system could span across hundreds of atoms, and this poses a significant challenge for CCSD(T) due to its steep computational scaling with both system and basis set size, severely limiting its applicability for realistic surface chemistry.

This work considers advances that address the challenges of simulating realistic material surfaces with these quantum many-body methods, as well as using these developments to validate the accuracy and enable key insights into a chemically diverse range of critical surface adsorption problems. This advance in simulation capabilities leverages the growing field of quantum embedding to introduce a controllable locality approximation that achieves a practical linear scaling in computational effort for system sizes considered[24]. Specifically, we extend the previously proposed 'systematically improvable quantum embedding' (SIE) method[25–27], which itself builds on density matrix embedding theory and fragmentation methods of quantum chemistry[21,28–30]. Our developments extend the approach to couple

together layers of different resolutions of correlated effects at different length scales, up to the 'gold standard' CCSD(T) level. We demonstrate exceptional accuracy with linear scaling, crucial for performing these simulations at the required scale. We also adapt the approach to utilize graphics processing units (GPUs) to eliminate key computational bottlenecks in our workflow, including the implementation of GPU-enhanced correlated solvers. With these, we can converge unprecedented simulations at the CCSD(T) level over solid-state systems of tens of thousands of orbitals, as summarized by the workflow of Fig. 1a and the observed computational scaling of Fig. 1b, with more details of these methodological advances described in Section "Methods" and Supporting Information (SI) Section S1.

## Results

### Water molecule on graphene

The adsorption of water on surfaces plays a central role in fields from desalination[31] to clean energy[32,33] and many more besides. The case of graphene represents a remarkably fundamental surface for this process, with its semi-metallic native state harboring a number of novel properties and promising applications including blue energy[33] and quantum friction[34]. However, the dominating weak and long-range van der Waals interactions between water and graphene lead to a highly non-local interaction that poses significant technical challenges to

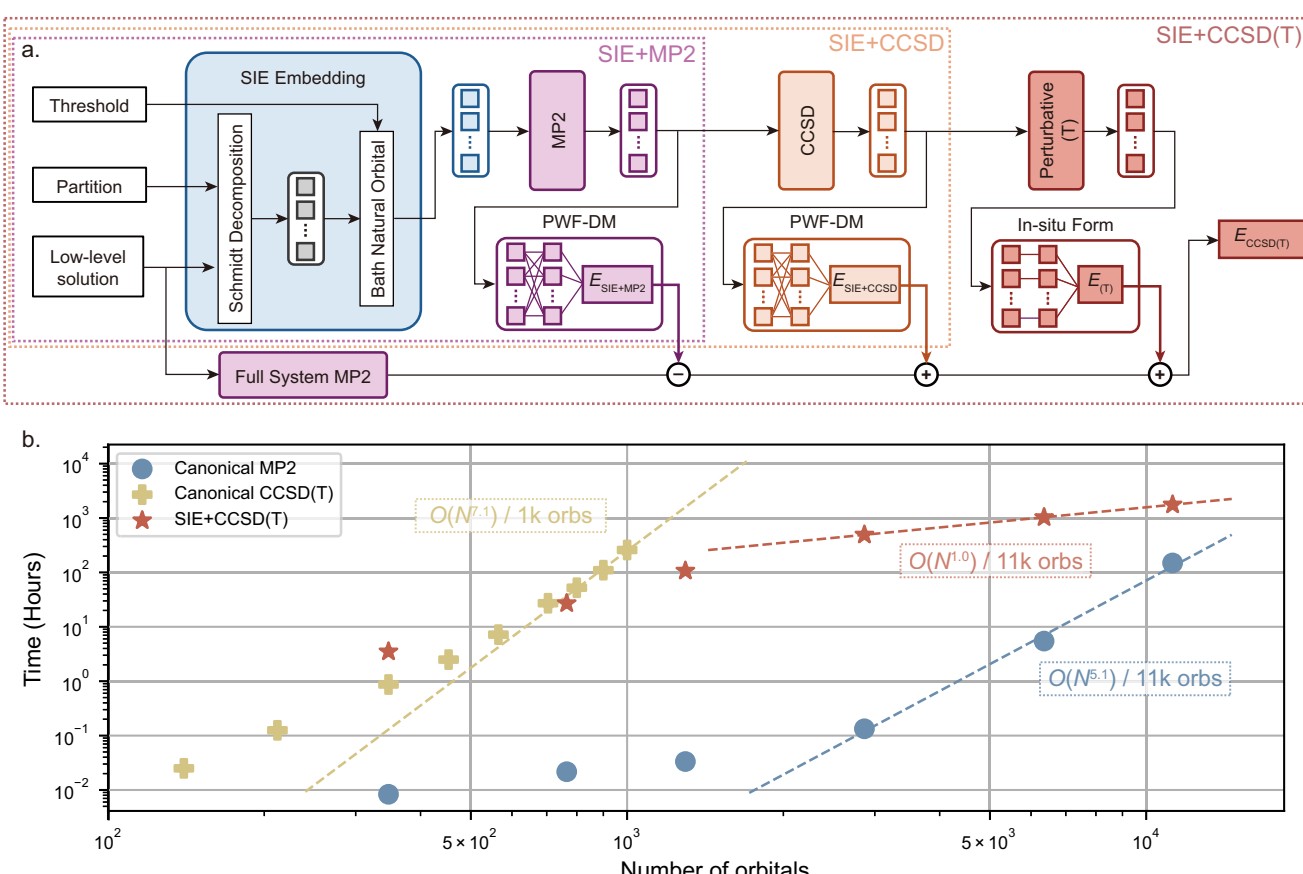

**Fig. 1 | Systematically improvable quantum embedding (SIE) framework and our Graphics Processing Unit (GPU)-accelerated quantum embedding for surfaces. a** illustrates the general workflow for a controllable multi-scale resolution of correlations at different length scales, comprising SIE with Møller-Plesset second-order perturbation theory (SIE+MP2), SIE with Coupled Cluster Singles and Doubles (SIE+CCSD), and with extra perturbative Triples (SIE+CCSD(T)). Specifically, the energy output layer of SIE+MP2 and SIE+CCSD employs the partitioned wave function density matrix method (PWF-DM), while SIE+CCSD(T) further adopts the in-situ form for perturbative (T). Further details in the Methods Section and

supplemental information (SI) Section S1. **b** presents the computational time as a function of the total number of orbitals, measured on a single A100 GPU for systems comprising water monomer adsorbed on graphene under open boundary condition. The annotations beside the curves show the scaling obtained by fitting the last three points up to the maximum size of the test system. The dashed line corresponds to a linear fit to the last three data points in each method. Further discussion on the observed linear scaling of SIE+CCSD(T) are provided in SI Section S1.6.

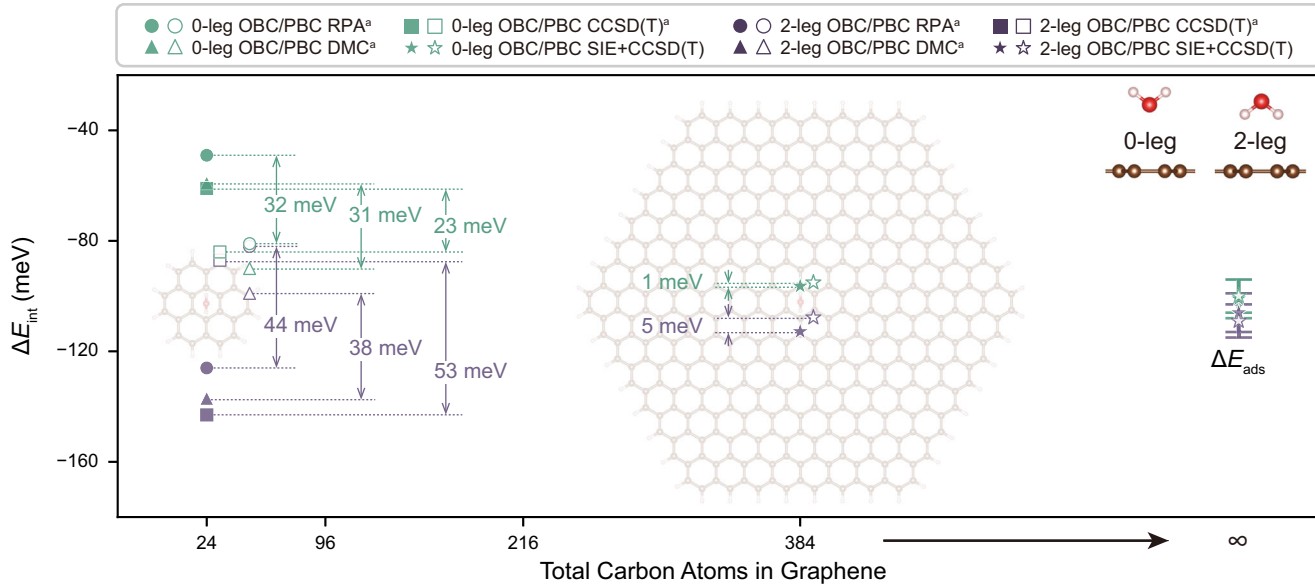

**Fig. 2 | The adsorption energy of water on graphene at the bulk limit.** The computed interaction energies ($\Delta E_{int}$) of $H_2O$@graphene are shown for both 0-leg and 2-leg configurations as well as with open and periodic boundary conditions. Side views of 0-leg and 2-leg configurations are shown in the top right of the figure. Left side displays previous results calculated by Random Phase Approximation (RPA), Diffusion Monte Carlo (DMC) and CCSD(T) from ref. 39 (superscript a) with the method and the OBC-PBC gap given, with the backdrop showing the largest OBC system ($H_2O$@PAH(2)). The middle of the figure shows the SIE+CCSD(T) results with the backdrop showing our largest OBC model ($H_2O$@PAH(8)). The right side presents the final adsorption energies, $\Delta E_{ads}$, obtained by considering both bulk limit extrapolation and geometry relaxation. The infinity symbol on the x-axis represents an extrapolation to the bulk limit. The details for the bulk limit extrapolation can be found in SI Section S3.8 and for geometry relaxation in SI Section S3.9. The SIE+CCSD(T) calculations are performed using cc-pV(D,T)Z extrapolated complete basis set with the neon-core correlation consistent effective core potentials[77,78].

**Table 1 | The final adsorption energies, $\Delta E_{ads}$, obtained by considering both bulk limit extrapolation and geometry relaxation**

| $\Delta E_{ads}$ (meV) | 0-leg | 2-leg |
|---|---|---|
| OBC | −101 ± 7 | −106 ± 7 |
| PBC | −100 ± 6 | −109 ± 6 |

achieve convergence with respect to the size of the graphene sheet[35-44]. Quantum many-body methods have so far been applied only to very limited graphene size[38,39]. While DFT can access much larger models[41,42] and the more efficient density functional tight binding approach can even process dynamical simulations of the water-graphitic interface[43,44], their accuracy is sensitive to the choice of underlying functionals. A method capable of reliably computing the adsorption energies of water on graphene ($H_2O$@graphene) therefore serves as a meaningful indicator of the feasibility of simulating realistic surface chemistry more broadly.

This limiting finite-size error can be qualitatively estimated via the difference between adsorption energies calculated for structures of similar size under both open and periodic boundary conditions (OBC and PBC, respectively), regarded as the OBC-PBC gap[39]. The OBC-PBC gap arises because the OBC and PBC models exhibit a different convergence of their finite-size errors, stemming from their contrasting physical origins. In OBC models, the error arises from the artificially truncated range of the interactions of the finite-sized substrate with the adsorbate, whereas under PBC conditions it is caused by the spurious periodic interactions between all the particles with their images in neighboring cells. Our developments provide an opportunity to confidently eliminate the finite-size error via a handshake between the adsorption energies computed under these different boundary conditions.

We first consider two configurations of the water-graphene adsorption system, namely the 0-leg and the 2-leg configurations, as shown in the upper right of Fig. 2, with the substrate size systematically enlarged. For OBC models, we extend the substrate up to $C_{384}H_{48}$, which is a hexagonal-shaped polycyclic aromatic hydrocarbon structure with a formula $C_{6h^2}H_{6h}$ ($h = 8$), also referred to as PAH(8), as shown in the background of Fig. 2. For PBC models, the substrate is extended up to a 14 × 14 supercell of 392 carbon atoms, which is the closest in size to PAH(8). Both the largest OBC and PBC systems contain more than 11,000 orbitals, at which point the OBC-PBC gap is reduced to 5 meV (1 meV) for 2-leg configuration (0-leg configuration). The gap of 2-leg configuration (0-leg configuration) is further narrowed to 3 meV (below 1 meV) by considering the effects of geometry relaxation and bulk limit extrapolation in adsorption energies, given in the Table 1. These small OBC-PBC gaps suggest that our results are effectively free of finite-size errors, an important aspect that was absent in previous computational studies of correlated methods for this system, where significant OBC-PBC gaps persisted[39]. With bulk limit extrapolations, our calculations also demonstrate that the interaction range for 0- and 2-leg water adsorption extends over distances exceeding 18 Å, which requires 400 carbon atoms in the computational models. As shown in Fig. 3a and Fig. S7 in SI, the interaction energies converge quite slowly as a function of the graphene size, which explains the large OBC-PBC gaps observed in previous works, as illustrated on the left side of the Fig. 2, where high-level ab-initio calculations were employed on systems within only 50 carbon atoms[39].

The 0- and 2-leg configurations are unique cases in which the dipole moment of the water molecule is perpendicular to the graphene surface. Water adsorption on graphene with varying orientations has not been extensively explored, but represents an important problem to address to understand the dynamics of water across graphene and subsequent implications in tribology applications[34,42,45]. We can characterize the orientation of the water by an angle of rotation, $\theta$, with the

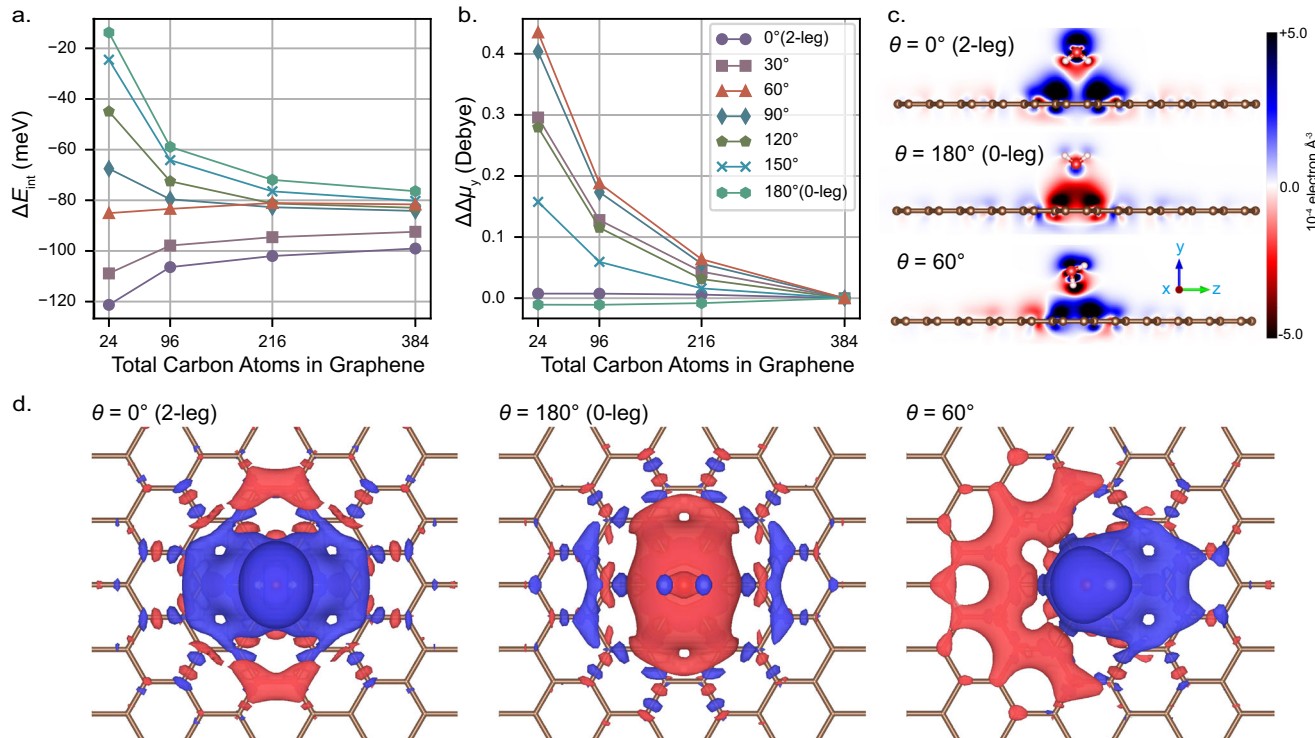

**Fig. 3 | Adsorption of water on graphene at different orientations. a** Interaction energy of water adsorption on graphene at various orientation angles. Calculations are performed at the SIE+CCSD level under OBC using PAH structures ranging from PAH(2) to PAH(8). **b** Adsorption-induced change in the dipole moment along the axis out of the plane of the graphene substrate (y-axis). To highlight the relative trends, all curves are shifted so that their values coincide for the 384 carbon graphene system. **c** The cross-sectional views of the electron density rearrangement distribution for PAH(6) with different water orientations. **d** Top views of the isosurface of electron density rearrangement, plotted with a value of $6.0 \times 10^{-5}$ (blue) and $-6.0 \times 10^{-5}$ (red) electrons Å$^{-3}$. The SIE+CCSD calculations are performed using cc-pVDZ basis set.

2-leg configuration defined at $\theta = 0°$ and the 0-leg configuration defined at $\theta = 180°$, with additional computational details in the SI Section S3.5.SIE+CCSD calculations were conducted on PAH(h) under OBC with $h = 2, 4, 6, 8$, and the results are shown in Fig. 3a. Again, we observe significant finite-size effects which change both the relative ordering and absolute scales of these adsorption energies over the different orientations. In particular, we find that the long-range interaction depends significantly on the orientation of water and is maximized for the 0-leg and 2-leg configurations. Furthermore, long-range interaction stabilize the adsorption for $\theta > 60°$, and destabilizes the adsorption for $\theta < 60°$. This emphasizes the importance of being able to converge to the bulk limit for all orientations to correctly describe their relative stability. Notably, for the configuration at $\theta = 60°$, the interaction energy remains nearly constant as the graphene substrate size increases, indicating that finite-size errors are particularly small in this specific case.

For more insight into these observations, we analyze the adsorption-induced dipole moment (see the definition in SI Section S3.13). Unlike the interaction energy in Fig. 3a, the adsorption-induced dipole moment for the $\theta = 60°$ configuration exhibits pronounced changes as a function of substrate size, as shown in Fig. 3b. This clearly shows that the seemingly short-ranged locality of the interaction energy in the $\theta = 60°$ configuration is merely an artifact of error cancellations.

To visualize this, we obtain the adsorption-induced electron density rearrangement (see definition in SI Section S3.12)[39] over the interaction range, for the configurations of 2-leg, 0-leg, and $\theta = 60°$, with their cross-sectional side views shown in Fig. 3c, and isosurface top views in Fig. 3d, respectively. For the 2-leg configuration, the dipole of the water points toward the substrate, pulling electrons from the graphene to the adsorption site and enhancing the water-graphene

interaction, as shown in Fig. 3c, d. The electron density rearrangement around the adsorbed site converges to the bulk limit faster than the free surface, thereby weakening the interaction energy as the system size increases to the converged limit for $\theta < 60°$.

For the 0-leg configuration, electrons in the substrate are pushed away from the adsorption site. Larger substrates allow for the repelled electrons to more effectively rearrange and relax, thereby lowering the energy of the adsorbed system and strengthening the interaction towards bulk limits for $\theta > 60°$.

For the special case of $\theta = 60°$, a balance is reached between these two effects as electrons are pulled from one side and pushed to the other side, as shown in Fig. 3d. This balance largely cancels out the finite-size effects of the interaction energy. However, this does not mean that the effects of the interaction for $\theta = 60°$ are particularly short-ranged. As can be seen from the electron density rearrangement in Fig. 3c, d, the extent of density changes in the $\theta = 60°$ system is not smaller than that for the 0-leg and 2-leg, and this cancellation for rapid convergence of the interaction energy with system size is likely to be only found for energetic properties. Based on this we can conclude that a full description of the interaction-driven effects between water and graphene are all long-ranged.

The substantial differences in finite-size convergence for these rotations underline the importance of being able to rigorously converge the size of the graphene substrate before conclusions can be drawn. Considering that the extrapolated SIE+CCSD(T) adsorption energies for the 0-leg and 2-leg configurations are already within their associated uncertainties of each other, it is expected that the adsorption energies among all configurations could be comparably close. This means that graphene does not exhibit a strong preference for any specific water orientation. Previous studies[38,39] also observed that the 0-leg, 1-leg, and 2-leg configurations exhibit similar adsorption

energies, but due to the limited number of sampled configurations, they were unable to reach such a general conclusion. This phenomenon is unusual for water, as the polar and directional hydrogen-bonding nature of water molecules usually results in a pronounced preference for specific adsorption configurations on surfaces. This behavior suggests that water-graphene interaction is dominated by non-directional van der Waals forces, unlike the interaction between water and other surfaces[46–49], such as the hexagonal boron nitride surface[46] or two–dimensional transition–metal dichalcogenide surfaces[48]. This is also supported by weak interaction analysis[50–52] presented in SI Section S3.14.

The interaction between water and a graphitic surface gives rise to many intriguing but not yet fully understood phenomena. Recent studies found that the friction for a water flow decreases as the diameter of carbon nanotubes decreases, a phenomenon known as quantum friction[34,42,45]. The water-graphitic interface gives rise to anomalous dielectric response[53,54]. It indicates that the standing descriptions of water at these interfaces based on lower-level theories are insufficient. Our results further highlight the necessity to pursue a high-level theoretical description of the water-graphene interface to explain these counter-intuitive emergent phenomena.

### Carbonaceous molecules on various surfaces

Carbonaceous molecules are ubiquitous in nature and synthetic materials. The adsorption of these carbonaceous molecules plays a crucial role in various fields, such as pollution control, drug design, and catalyst optimization[55–59]. As a key quantity in determining surface chemistry, adsorption energies have been computationally probed for various carbonaceous molecules and surfaces to understand their behavior[47,60–73]. To demonstrate the wide applicability of our method, we consider three representative systems with chemically diverse surfaces selected: carbon monoxide on the MgO(001) surface, denoted as 'CO@MgO', six organic molecules on coronene, a simplified model to mimic graphene, denoted as 'Organic molecules@Coronene'[60], and carbon monoxide[63]/carbon dioxide[64,65] on the metal-organic framework CPO-27-Mg, denoted as 'CO/$CO_2$@CPO-27-Mg'. Fig. 4a−c shows the adsorption configurations and the surfaces/clusters. The adsorption energies across these systems range from 4 to 14 kcal mol⁻¹. This range spans from typical weak physisorption to strong chemisorption, highlighting the diversity of these distinct surface types.

To demonstrate the robustness and versatility of the SIE+CCSD(T) protocol, we compare our results to a variety of computational approaches across the systems. Detailed descriptions are provided in the SI Section S4, and we present only the main results here. The computed values, compared to their corresponding experimental data, are shown in Fig. 4d, where the black line represents perfect agreement between theory and experiment, and the gray shade indicates deviation within a 'chemical accuracy' of ±1 kcal mol⁻¹. We note that the accuracy of the same method, for example MP2, can vary among different adsorption systems. Furthermore, even in the same system (such as CO@MgO), different implementations can result in significant differences, due to the convergence of technical parameters. Notably, the SIE+CCSD(T) method achieves a sub-chemical accuracy in agreement with experimental data across all studied systems.

The inset in Fig. 4d further illustrates the deviations of SIE+MP2, SIE+CCSD, and SIE+CCSD(T) relative to experimental values, highlighting the significant improvement in accuracy of SIE+CCSD(T) over SIE+MP2 and SIE+CCSD. Another important reason is that SIE+CCSD(T) maintains the size consistency inherent in Coupled Cluster theory (discussed in SI Section S4.2), which is crucial for achieving consistent high accuracy across systems of different sizes. A contrasting example is MP2, which is widely known as lacking size consistency. Although it performs very well in systems involving the adsorption of small molecules, such as CO/$CO_2$@CPO-27-Mg and CO@MgO, in the case of Organic molecules@Coronene, where all six

types of organic molecules are larger than the CO/$CO_2$ examples, MP2 performs poorly. As described in SI Section S4.2, as the size of organic molecules increases, the differences between MP2 and experimental values significantly widen, whereas the differences between SIE+CCSD(T) and experimental values do not show as marked a change. This consistent high-accuracy performance bolsters our confidence that SIE+CCSD(T) possesses good versatility and transferability to be applied across complex surfaces and with a wide variety of adsorbates.

## Discussion

High-accuracy simulations of realistic material surfaces using quantum many-body methods inevitably encounter two main challenges: (1) accurately capturing the essential electron correlations within materials, and (2) favorable computational scaling to reach the bulk limit, which can involve hundreds of atoms and tens of thousands of basis functions or more to account for long-range interactions. In this work, we demonstrate that our highly efficient GPU-accelerated SIE+CCSD(T) framework effectively addresses these challenges for typical monomer adsorption systems. For water on graphene − a famously challenging system due to its pronounced long-range interactions − we scaled our calculations to converge to CCSD(T)-level accuracy through a systematic computational approach. This allowed us to eliminate finite-size errors which stymied previous studies. At the bulk limit, the adsorption energy converges to approximately 100 meV for both the 0-leg and 2-leg configurations, both lying inside the uncertainty of the calculations. The interaction energies for different orientations of water are also within this range, suggesting that graphene is insensitive to the orientation of water monomer adsorbed on its surface, contrary to previous findings. For structurally more complex systems of carbonaceous molecules on chemically diverse surfaces, our predictions for adsorption energies are within chemical accuracy of experimental values, demonstrating a good versatility and accuracy of SIE+CCSD(T). For a broader range of scenarios such as cluster adsorption on surfaces, surface coverage by adsorbates at finite densities, or surface doping, SIE+CCSD(T) does not encounter any workable limitations, thus enabling its application in tackling a wide array of material surface problems.

This large scale of quantum chemistry simulations for material surfaces has been enabled by our newly developed SIE+CCSD(T) framework, which leverages the linear scaling nature of fragmentation approaches alongside the computational power of modern GPUs. This efficient framework sets the stage for understanding the broader mechanisms of molecular surface chemistry. Integrating SIE with solvers that can assess excited states will allow for the accurate modeling of photochemistry at surfaces and non-radiative transitions in surface defects and quantum dots. Furthermore, combining this framework with reaction path searching tools will enable detailed mapping of surface reaction processes, and potentially offer predictive understanding of complex chemical reactions on surfaces.

## Methods

### Framework of SIE+CCSD(T)

Quantum embedding methods strike a balance between accuracy and computational simplicity by selecting a set of overlapping *clusters* from the system for a high-level numerical method treatment (such as FCI, CCSD(T), QMC, etc), integrated into a low-level (e.g., mean-field) representation of the surrounding *environment*. These clusters are comprised of local atomic-like fragment spaces over each atomic species, along with a *bath* space to span the physics of quantum fluctuations from the fragment into its environment as described by the low-level theory. However, the intrinsic mismatch between low-level and high-level solutions in reconstructing the total energy can lead to a lack of clear improvability in quantum embedding, making it difficult to systematically improve. SIE provides a more reliable route towards convergence in energetic quantities by expanding the bath space with

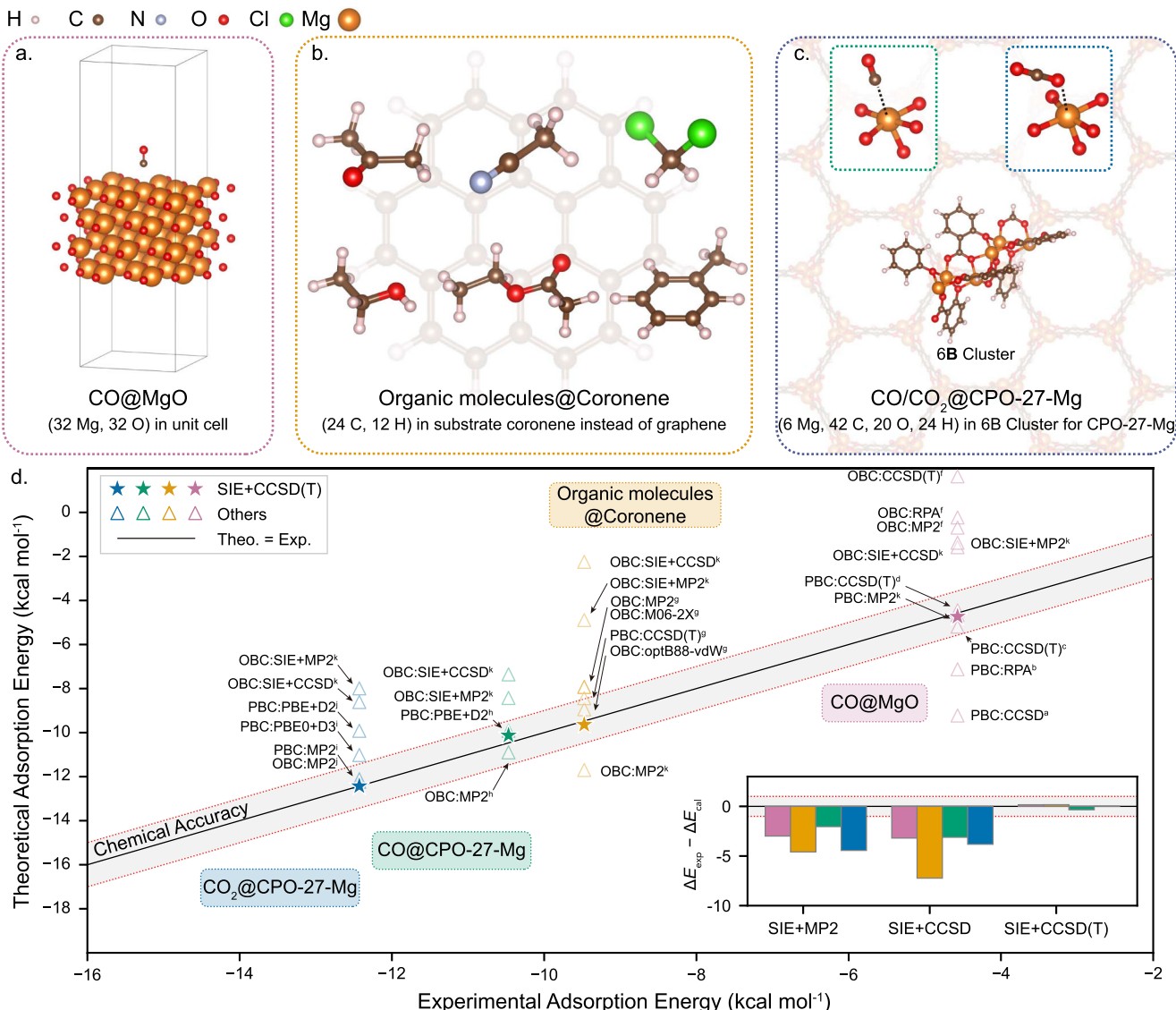

**Fig. 4 | Structures and adsorption energies for carbonaceous molecules on different systems. a** CO on a 4-layer MgO(001) surface with PBC. **b** 6 organic molecules used as the adsorbate on graphene which is replaced by Coronene under OBC. **c** CO/$CO_2$ adsorption configurations in metal-organic framework CPO-27-Mg. The 6**B** cluster is cut from CPO-27-Mg, used to calculate interacting energies under OBC. In **b**, **c** to clearly show all adsorbates, their substrates have been made transparent and displayed as a background. Structures are rendered by using VESTA[79]. **d** Adsorption energies obtained through SIE+CCSD(T) and other methods. Among the methods, Perdew-Burke-Ernzerhof (PBE) is a generalized gradient approximation (GGA) functional[80], while PBE0 is a hybrid functional[81], M06-2X is a meta-GGA functional[82], and optB88-vdW is a van der Waals density functional[83,84]. D2[85] and D3[86] are empirical dispersion interaction corrections. An inset shows differences between SIE+MP2/CCSD/CCSD(T) and experimental data. The references are labeled as superscript from (a-j) corresponding to published data from refs. 60,61,63−65,67,87−90, respectively, while k refers to this work. The shaded area between the dotted red lines in (**d**) represents a difference of ±1 kcal mol⁻¹, corresponding to chemical accuracy.

carefully designed bath natural orbitals (BNOs) relying on a perturbative description of these fluctuations[25].

The SIE framework, outlined in Fig. 1a, requires three key components: a low-level solution, precomputed using mean-field theory (such as the Hartree-Fock method employed in this work); a partitioning strategy designed to generate clusters for quantum embedding; and a BNO threshold to decide the number of BNOs at each level of theory in which the clusters are described. The SIE embedding procedure begins by forming a mean-field bath through the use of the Schmidt decomposition, as employed in density matrix embedding theory (DMET)[28,29]. The BNOs augment this bath space based on the entanglement between the fragment and its surrounding environment obtained through second-order Møller-Plesset perturbation theory (MP2) over the full system orbitals. Only BNOs with entanglement greater than the threshold are included in the cluster at each level of theory. For a detailed description of the SIE implementation, including the Schmidt decomposition and the BNO building, see SI Section S1.

Following this, the many-body problem over each cluster is solved at their respective level of theory, first undergoing MP2, then CCSD on top of the MP2 results, and then the final CCSD(T) calculation. After merging, the high–level solutions of clusters at each hierarchical tier are assembled into full system result at the corresponding level of theory, thereby forming the SIE+MP2, SIE+CCSD, and SIE+CCSD(T) schemes, which are distinguished by different colors in Fig. 1a. It is important to note that although MP2 and CCSD can be used directly as solvers, CCSD(T) requires directly downfolding correlation into fragments. Therefore, CCSD(T) in SIE is actually a composite approach. A detailed description of perturbative (T) in SIE can be found in the SI Section S1.3.

We employ the partition wavefunction density matrix (PWF–DM) approach introduced in ref. 26 to merge the high-level solutions of all clusters and compute the total energy or other observables' expectations. PWF–DM constructs the full-system reduced density matrix (RDM) directly from the individual cluster wavefunction and evaluates the total energy using this full-system RDM. Because the RDM is derived explicitly from wavefunctions, the resulting quantities are rigorously $N$–representable. Moreover, by including cross–cluster interaction during the assembly of the full–system RDM, PWF–DM attains higher accuracy and improves the convergence of energies with respect to cluster size than the method that relies solely on in–cluster energy calculations. Implementation details of the PWF–DM approach are given in SI Section S1.2. PWF–DM can be applied to cluster merging in both SIE+MP2 and SIE+CCSD at negligible additional cost. For the perturbative (T) contribution, we introduce an RDM formulation that includes cross–cluster interaction, referred to as the ex–situ form for perturbative (T). Further theoretical description is provided in SI Section S1.3.2. However, the ex-situ form may potentially increase the scaling of SIE. Therefore, we instead in-cluster evaluate the perturbative (T) correction by using the in-situ form, whose details are documented in SI Section S1.3.1.

Finally, due to the cutoff in the BNO space, some weakly entangled bath orbitals are excluded, which leads to the loss of a small portion of the long-range correlation for a given fragment. This bath truncation error caused by the incompleteness of the cluster space can be quantified and corrected at the MP2 level via the difference between a full–system MP2 calculation and the corresponding SIE+MP2 result[74]. The full system MP2 is not the only candidate to capture this error. Any method, which could estimate the correlation in a larger cluster space than SIE+MP2, can do the same thing, such as a SIE+MP2 with a lower BNO threshold.

Therefore, as depicted in Fig. 1a, the SIE+CCSD(T) total energy $E_{CCSD(T)}$ can be computed as

$$E_{CCSD(T)} = E_{mf} + E(\Gamma_{CCSD}) + E(C_{(T)}) + E_{MP2}^{Full} - E(\Gamma_{MP2}),  \quad (1)$$

where $E_{mf}$ is the mean-field energy obtained from the low-level solution, $E(\Gamma_{MP2})$ and $E(\Gamma_{CCSD})$ are the correlation energies from SIE+MP2 and SIE+CCSD obtained via PWF–DM, respectively, and $E(C_{(T)})$ is the correlation energy obtained from in-situ form perturbative (T). $E_{MP2}^{Full}$ is the MP2 energy of the full system, and $E_{MP2}^{Full} - E(\Gamma_{MP2})$ serves as the correction for the bath truncation error.

Additionally, we have implemented a GPU-accelerated version of this quantum embedding workflow to dramatically increase computational throughput. Some details for GPU implementation can be found in SI Section S2. As shown in Fig. 1b, the observed complexity of the method has been fitted. Due to noticeable inflection points for those methods in larger systems, only the last three data points were selected for fitting. Interestingly, we observed a linear scaling for SIE+CCSD(T), indicating that with effective engineering optimizations, theoretical bottlenecks do not dominate practical computations for system sizes up to 11,423 orbitals in SIE+CCSD(T). For further discussion on the linear scaling and computationally intensive steps of SIE+CCSD(T), we refer to SI Section S1.6. Specific settings for SIE, including the selection of BNOs, partition schemes, choice of basis sets, as well as detailed error estimation and error correction, can be found in Section S3–S5.

## Adsorption energy

The adsorption energy is defined as

$$\Delta E_{ads} = E(AB) - E(A) - E(B),  \quad (2)$$

where $\Delta E_{ads}$ represents the adsorption energy, $E(AB)$ is the total energy of the system A adsorbed on B, and $E(A)$ and $E(B)$ are the total energies of A and B in their isolated thermal equilibrium structures, respectively. In practice, the adsorption energy can be decomposed into two components:

$$\Delta E_{ads} = \Delta E_{int} + \Delta E_{geom},  \quad (3)$$

where $\Delta E_{int}$ denotes the energy arising from the interaction between A and B in their equilibrium geometries, and $\Delta E_{geom}$ is the energy resulting from geometry relaxation of the adsorbate relative to its isolated equilibrium structure[61]. Generally, $\Delta E_{geom}$ is very small for surface adsorption and can thus be estimated using DFT calculation as a correction to the adsorption energy. For more details, we refer to Section S3.9 of the SI. The interaction energy, which constitutes the major part of $\Delta E_{ads}$, requires accurate computation. However, for the method using Gaussian type orbitals (GTO) as basis sets, the basis set superposition error (BSSE) can arise in adsorption energy calculations[75]. A common approach to address the BSSE is counterpoise correction, applied to $\Delta E_{int}$ using equation

$$\Delta E_{int} = E(AB) - E(A[B]) - E([A]B),  \quad (4)$$

where $E(A[B])$ refers to the energy of the AB geometry, calculated with only the basis functions of B included, while the electrons and nuclei of B are excluded, and similarly for [A] in $E([A]B)$.

### Reporting summary

Further information on research design is available in the Nature Portfolio Reporting Summary linked to this article.

## Data availability

All data generated in this study are provided in the Supplementary Information and Source Data file. All geometries used in this study can be found in the Supplementary files. Source data are provided with this paper.

## Code availability

We opensource the implementation of SIE+CCSD(T) (https://github.com/bytedance/byteqc) on GitHub[76] with the license of Apache-2.0.

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

## Acknowledgements

We extend our gratitude to Dr. Hang Li and ByteDance Research for their invaluable support. We also thank Dr. Hongzhou Ye for his insightful discussions on the project. We appreciate Dr. Kaido Sillar and Dr. Joachim Sauer for generously sharing the CPO-27-Mg structure. Additionally, we thank Ruichen Li, Weiluo Ren, Weizhong Fu and Qingyuan Zhang for their suggestions to improve the paper's quality. J.C. is supported by the National Key R&D Program of China under Grant No. 2021YFA1400500 and the National Science Foundation of China under Grant No. 12334003.

## Author contributions

Z.H. and D.L. conceived the study; Z.H. implemented the main code with contributions from Z.G.; X.W., J.C., D.L., and G.H.B. suggested the experiments; Z.H. performed simulations and analyzed the results; X.W., C.C., J.C., H.Q.P., and G.H.B. performed the chemical or physical analyses; Z.H., D.L., H.Q.P., J.C., X.W., and C.C. performed figure designing; Z.H. and D.L. organized the project; Z.H., C.C., H.Q.P., X.W., G.H.B., J.C., and D.L. wrote the paper.

## Competing interests

The authors declare no competing interests.
