## [Transparent Peer Review file · Nature Communications]

A multi-resolution systematically improvable quantum embedding scheme for large-scale surface chemistry calculations

Corresponding Author: Dr Zigeng Huang

Version 0:

Reviewer comments:

Reviewer #1

(Remarks to the Author)

Huang et al. present an approach based on the systematically improvable quantum embedding (SIE) method to address problems in surface chemistry. The methodology is developed and applied at the SIE+CCSD(T) level to water on graphene surfaces to determine differences in binding energies with orientation, as well as other small carbonaceous molecules on various surfaces. The calculations performed here achieve impressive scaling with system size while also providing insight into the strength and nature of these binding interactions, in particular reaching sub-chemical accuracy in agreement with experimental data for the carbonaceous systems.

While this work introduces an interesting methodology and applies it to several chemical systems, we suggest that the work would be a more appropriate fit in another journal, such as the Journal of Chemical Theory and Computation or the Journal of Physical Chemistry C with several edits. As currently submitted, the paper may not adequately describe the novel methodological developments, and the systems evaluated may not be of appropriately broad interest. Broadly, we would suggest that this paper would be more strongly received by focusing on the methodological work and advancements in further depth in the main text, rather than the Supporting Information.

(Remarks on code availability)

See Attached

Reviewer #2

(Remarks to the Author)

See attached review

(Remarks on code availability)

Reviewer #3

(Remarks to the Author)

In this manuscript, the authors extended the "systematically improvable quantum embedding" (SIE) approach to achieve CCSD(T) level accuracy for the binding energy between molecules and substrates. Given the importance of energetics in surface science (such as catalysis), this work will open new doors for accurate determinations of binding geometries and binding energies and will be of broad interest. The paper is nicely written, with sufficient technical details in the Supporting Information. I recommend publication in Nature Communications, with the following comments for the authors to consider.

(1) On page 9 of the main text, at the beginning, the authors mentioned that the geometry relaxation effect is typically small

and can be captured at the DFT level. I am not sure I fully understand this. There are two related questions: (i) Can the authors justify this statement based on some explicit calculations, even for a very small system (I understand that the geometry optimization is expensive with SIE, as the authors mentioned in Sec. 2.8 of the SI). (ii) When one uses DFT to capture this effect, what is the recommended procedure or functional? In Sec. 2.8 of the SI, the authors mentioned that B97M-V and omegaB97M-V are used for the systems studied in this work: does this conclusion apply generally to other systems, or is there a recommended procedure for finding the "optimal" functional?

(2) It appears to me that in all the systems studied in this work, there is a single molecule adsorbed on a large-area substrate surface, and one needs to check the convergence to the "bulk" limit (where the surface is infinitely large). However, a more common experimental scenario is that, there is a finite coverage, i.e., one adsorbed molecule per $X \text{ nm}^2$ (X is a number that depends on the experimental condition). The coverage can be very low in some cases, but the "bulk" limit discussed in this work does not seem to be physical. The authors should comment on this point, and should also comment on whether this approach can be extended to cases to accommodate the finite coverage (presumably, treating the molecular adsorbates as periodic images).

(3) Throughout the manuscript, there are a few places where "adsorption" was spelled incorrectly as "absorption" and should be corrected.

(Remarks on code availability)

Version 1:

Reviewer comments:

Reviewer #2

(Remarks to the Author)

I have been asked to comment on the authors' responses to both Reviewer 1 and my own. The authors have given a thorough response to Reviewer 1's comments (as well as my own). This manuscript is novel and should be published.

(Remarks on code availability)

Reviewer #3

(Remarks to the Author)

In this revision, the authors have satisfactorily addressed all my concerns from the previous round of review. The readability and clarity improved greatly. Therefore, I recommend publication as is.

(Remarks on code availability)

Huang et al. present an approach based on the systematically improvable quantum embedding (SIE) method to address problems in surface chemistry. The methodology is developed and applied at the SIE+CCSD(T) level to water on graphene surfaces to determine differences in binding energies with orientation, as well as other small carbonaceous molecules on various surfaces. The calculations performed here achieve impressive scaling with system size while also providing insight into the strength and nature of these binding interactions, in particular reaching sub-chemical accuracy in agreement with experimental data for the carbonaceous systems.

While this work introduces an interesting methodology and applies it to several chemical systems, we suggest that the work would be a more appropriate fit in another journal, such as the Journal of Chemical Theory and Computation or the Journal of Physical Chemistry C with several edits. As currently submitted, the paper may not adequately describe the novel methodological developments, and the systems evaluated may not be of appropriately broad interest. Broadly, we would suggest that this paper would be more strongly received by focusing on the methodological work and advancements in further depth in the main text, rather than the Supporting Information.

We have specifically separated our major comments for this work towards targeting each of these journals.

If the authors prefer to submit the work to Journal of Chemical Theory and Computation, we suggest following comments to guide the more technically-inclined readers:

1. Authors present a workflow that extrapolates both the open boundary condition (OBC) and periodic boundary condition (PBC) calculations to the thermodynamic limit separately to assess the finite-size errors. We ask that the authors consider explain the methodology in more detail. For example, we find the “handshake” between the OBC and PBC adsorption energies interesting and would like to see more details. We would also like to see how the differences in the bath natural orbitals (BNOs) would interact with the extrapolations. The boundary conditions likely affects the automatic BNO construction in a distinct manner, notably due to the spurious interaction in the small supercell PBC regime. We believe that inclusion of further detail will help the readers fully understand the workflow.
2. The authors mention that they have “implemented a GPU-accelerated version of this quantum embedding workflow...”. Technically-oriented readers will likely appreciate further detail on how the GPU-acceleration is implemented. We believe that some worded description or figures would be beneficial.
3. The authors use “impurity” in the both the Methods section (L399) and Supporting Information (eg. Page 2). Clear definition of the terminology within the context of the author’s work would be appreciated by the readers.
4. The authors use PWF-DM as part of the proposed workflow. We are interested in learning why the authors made a particular choice among the other schemes introduced in the paper they cite (Nusspickel et.al **2023** JCTC). We are similarly interested in a further discussion into the energy calculation, as addressed in the Supporting Information and compared to the cumulant-based methods discussed in this work.
5. From what we understand, the perturbative (T) correction in SIE embedding is a novel technical contribution in this work. We are interested in seeing a benchmark for this new technique, especially as it appears that SIE+CCSD performs worse than

SIE+MP2 and perturbative (T) correction has a large impact. We also ask that the authors comment on their choice not to include the cross-cluster information in constructing the (T) correction.

6. In Figure 2, the pipeline for the calculations implies nested, additive corrections ...

If the authors would prefer to submit the work to the Journal of Physical Chemistry C, we suggest the following comments of interest to a materials-focused audience:

1. On page 5, the authors discuss the absorption-induced dipole moment. We would suggest that the authors consider including the absorption-induced dipole moment as a function of substrate size as a figure or subplot of a figure, as we believe that this could support the main claim that a full description of the interaction-driven effects between water and graphene are all long-ranged.
2. On the right side of page 5, the authors claim that "graphene does not exhibit a strong preference for any specific water orientation, which is unusual for water..." Based on what the authors specify in the Supporting Information about the CBS extrapolation of relatively limited double- and triple-zeta basis set calculations, it might be nice to include references to work in which water has exhibited an orientational preference to provide context and explain how this work is different.
3. On page 4, the authors state that "absorption induced electron density rearrangement is considered as another descriptor for interaction range." It may help the reader to provide references and a description of why electron density rearrangement is a good descriptor for interaction range.

We additionally offer the following minor suggestions, irrespective of which journal the authors prefer:

1. In Figure 1b, the authors present a computational time plot for their calculations. The choice of fitting the last three points suggest that some of the computational steps saturate the computational cost scaling at a sufficiently large enough system size. It would be valuable for the authors to provide a breakdown of the computational workflow and identify 1) the computationally intensive step and 2) its cost scaling for a clearer picture.
2. The authors introduce their partition strategy on page 15 of the SI. It seems like some fragments are rather small. For example, one edge fragment contains two hydrogen atoms. Does this introduce unnecessarily inexact fragment descriptions to the calculation workflow?
3. We request that the authors specify their choice of basis sets in the captions of Figures 2 and 3 to guide the readers.
4. Consider starting a new paragraph when beginning to discuss the 0-leg configuration on the left side of page 5.
5. The authors outline that they used Γ -point calculations only on Page 12 of the Supporting Information. Is k -point sampling a direction that the authors may find interesting to explore?

6. The authors discuss the BSSE errors and the counterpoise correction in Page 9. We are interested in reading if the authors made any modifications to the conventional counterpoise correction to account for the distinct *automatically* chosen bath natural orbitals.
7. Page 4 mentions tribology applications. The reader might be interested in slight elaboration on the relevance of this application, perhaps with a citation.
8. Paragraph 1 of page 3 provides numeric values for the OBC-PBC gaps for the 2-leg and 0-leg configurations on the left side of the page, and on the right side of the page provides differences between the OBC and PBC models for the 0-leg and 2-leg configurations. We would appreciate if the authors check for redundancy and consider restructuring this paragraph for clarity.
9. On the top of page 6 the authors state "SIE+CCSD(T) maintains the size consistency inherent in Coupled Cluster theory..." The reader may find systematic calculations that demonstrate size consistency useful.
10. We find Figure 2 to be a bit challenging to read: we would suggest making the figure more legible.
11. We would like to point out a few minor typos:
 - (a) (Page 8, Line 366) *sysetm MP2*
 - (b) (Page 8, Line 371) *Samething*

In this submission to Nat. Commun., the authors build upon state-of-the-art correlated wavefunctions to reliably converge to the 'gold standard' accuracy in quantum chemistry for application to extended surface chemistry. The authors use graphics processing unit acceleration along with systematically improvable multiscale resolution techniques to achieve linear computational scaling up to 392 atoms in size. The authors provide a new benchmark for this water-graphene interaction that clarifies the preference for water orientations at the graphene interface (further comments on this are given below). This is extended to the adsorption of carbonaceous molecules on chemically complex surfaces, including metal oxides and metal-organic frameworks. The authors conclude that their work enables more reliable and improvable approaches to first-principles modeling of surface problems at an unprecedented scale and accuracy using ab-initio quantum many-body methods.

I consider this work to be of interest to computational chemistry/materials researchers as well as readers of this journal. As such, I am generally supportive of publication with a few minor edits. The authors clearly have a capability that is superior to other "empirical" approaches such as DFT. However, there has been previous work in the field using DFTB to treat large systems, which should be noted: J. Phys. Chem. C 2016, 120, 19212–19224 and J. Am. Chem. Soc. 2024, 146, 35313–35320. Specifically, these prior studies also had the same goal of linear scaling of large solvated systems (although the treatments in these studies are not at the same level of accuracy of the work under study). In conclusion, I consider this work to be impressive, but the authors should also mention some of the historical development in this field of linear scaling for large solvated systems.

SUMMARY OF CHANGES

1. The Method section in the main text has been partly reorganised. We expanded the description of several SIE workflow components and added clear guidance regarding the position of the corresponding technical details in the SI.
 - Add the guidance on the practical Schmidt decomposition and the BNO building.

“ For a detailed description of the SIE implementation, including the Schmidt decomposition and the BNO building, see SI Sec. S1. ”
 - Refactor the descriptions for hierarchical tier of SIE+MP2, SIE+CCSD and SIE+CCSD(T)

“ After merging, the high-level solutions of clusters at each hierarchical tier are assembled into full system result at the corresponding level of theory, thereby forming the SIE+MP2, SIE+CCSD, and SIE+CCSD(T) schemes, which are distinguished by different colours in Fig. 1a. ”
 - Enrich the description of perturbative (T) and the partition wavefunction density matrix which serves as the merging method.

“ We employ the partition wavefunction density matrix (PWF-DM) approach introduced in Ref. [26] to merge the high-level solutions of all clusters and compute the total energy or other observables’ expectations. PWF-DM constructs the full-system reduced density matrix (RDM) directly from the individual cluster wavefunctions and evaluates the total energy using this full-system RDM. Because the RDM is derived explicitly from wavefunctions, the resulting quantities are rigorously N -representable. Moreover, by including cross-cluster interaction during the assembly of the full-system RDM, PWF-DM attains higher accuracy and improves the convergence of energies with respect to cluster size than the method that relies solely on in-cluster energy calculations. Implementation details of the PWF-DM approach are given in SI Sec. 1.2. PWF-DM can be applied to cluster merging in both SIE+MP2 and SIE+CCSD at negligible additional cost. For the perturbative (T) contribution, we introduce an RDM formulation that includes cross-cluster interaction, referred to as the ex-situ form for perturbative (T). Further theoretical description is provided in SI Sec. 1.3.2. However, the ex-situ form may potentially increase the scaling of SIE. Therefore, we instead in-cluster evaluate the perturbative (T) correction by using the in-situ form, whose details are documented in SI Sec. 1.3.1. ”
 - We modify the description of the bath truncation error and its correction.

“ Finally, due to the cutoff in the BNO space, some weakly entangled bath orbitals are excluded, which leads to the loss of a small portion of the long-range correlation for a given fragment. This bath truncation error caused by the incompleteness of the cluster space can be quantified and corrected at the MP2 level via the difference between a full-system MP2 calculation and the corresponding SIE+MP2 result [82]. ”
 - We move the total energy formulation from SI to the main text Method section.

“ Therefore, as depicted in Fig. 1a, the SIE+CCSD(T) total energy $E_{\text{CCSD(T)}}$ can be computed as

$$E_{\text{CCSD(T)}} = E_{\text{mf}} + E(\Gamma_{\text{CCSD}}) + E(C_{\text{(T)}}) + E_{\text{MP2}}^{\text{Full}} - E(\Gamma_{\text{MP2}}), \quad (1)$$

where E_{mf} is the mean-field energy obtained from the low-level solution, $E(\Gamma_{\text{MP2}})$ and $E(\Gamma_{\text{CCSD}})$ are the correlation energies from SIE+MP2 and SIE+CCSD obtained via PWF-DM, respectively, and $E(C_{\text{(T)}})$ is the correlation energy obtained from in-situ form perturbative (T). $E_{\text{MP2}}^{\text{Full}}$ is the MP2 energy of the full system, and $E_{\text{MP2}}^{\text{Full}} - E(\Gamma_{\text{MP2}})$ serves as the correction for the bath truncation error. ”

2. We have added Section S3.1, “Outline for the Corrections to Achieving OBC-PBC “Handshake” ”, to the SI, which outlines the details required for SIE+CCSD(T) for the water@graphene system to obtain this handshake between different boundary conditions.
3. We have added Section S2, “GPU-Accelerated High-Performance Package”, to the SI to outline the challenges we encountered when GPU-accelerating the software package and the solutions we adopted. A pointer directing readers to this has also been added in the Method section.
 “ Some details for GPU implementation could be found in SI Sec. S2.”
4. Several misuses of the term “impurity” have been corrected in the manuscript.
 - In the Method section of the main text it now is modified as “.....based on the entanglement between the fragmentimpurity and its surrounding environment.....”
 - In the SI Sec. 1.1, every occurrence of “impurity” has been replaced with “fragment+bath subspace”.
5. At the start of SI Sec. S1.2, where PWF-DM is introduced, we briefly summarize the four merging methods cited in [Nusspickel, Max et al., JCTC, 2023 19 2769] that can be employed in quantum embedding to form the fragment solutions into the full-system solution, and we explain in detail the reason for PWF-DM being chosen in this study.
6. We have added Sec. S1.3.2 in SI to describe, at the theoretical level, how cross-cluster information can be incorporated into the perturbative (T) calculation and discuss its feasibility.
7. We have added to Fig. 3 the curves showing the adsorption-induced dipole moment changes as graphene size increases, to prove that, for the $\theta = 60^\circ$ configuration, the interaction between the water monomer and graphene still remains long-ranged. The figure caption and the corresponding discussion in the main text have been updated accordingly.
 - For the caption:

“(b) Adsorption-induced change in the dipole moment along the axis out of the plane of the graphene substrate (y-axis). To highlight the relative trends, all curves are shifted so that their values coincide for the 384 carbon graphene system.”

- For the description in main text:

“Unlike the interaction energy in Fig. 3a, the adsorption-induced dipole moment for the $\theta = 60^\circ$ configuration exhibits pronounced changes as a function of substrate size, as shown in Fig. 3b. This clearly shows that the seemingly short-ranged locality of the interaction energy in the $\theta = 60^\circ$ configuration is merely an artifact of error cancellations.”

“To visualize this, we obtain the adsorption-induced electron density rearrangement (see definition in SI Sec. S3.12) [31] over the interaction range, for the configurations of 2-leg, 0-leg, and $\theta = 60^\circ$, ”

8. In the main text, we have added references to show that similar adsorption energies were observed only among a limited set of configurations, and therefore how we have extended this to clarify the insensitivity regarding the orientation of water on graphene. By comparison, water typically exhibits a strong orientation preference on other surfaces.

“This means that graphene does not exhibit a strong preference for any specific water orientation. Previous studies [31,41] also observed that the 0-leg, 1-leg, and 2-leg configurations exhibit similar adsorption energies, but due to the limited number of sampled configurations, they were unable to reach such a general conclusion. This phenomenon is unusual for water, as the polar and directional hydrogen-bonding nature of water molecules usually results in a pronounced preference for specific adsorption configurations on surfaces. This behavior suggests that water-graphene interaction is dominated by non-directional van der Waals forces, unlike the interaction between water and other surfaces [47-51], such as the hexagonal boron nitride surface [47] or two-dimensional transition-metal dichalcogenide surfaces [49].”

9. In the Method section of the main text, we have added a pointer directing readers to the discussion of the computationally intensive steps provided in the SI.

“For further discussion on the linear scaling and computationally intensive steps of SIE+CCSD(T), we refer to SI Sec. S1.6”

10. An explanation and discussion regarding the relatively small size of the fragments has been added to SI Section S3.3.

“ In this work, the fragments are kept deliberately small. This choice is made based on an insight that the BNO construction already embeds beyond-mean-field information from MP2, while the fragment and its bath are only constructed at the mean-field level. Expanding a cluster by adding BNOs is therefore markedly more efficient than enlarging the fragment itself. As discussed in the original SIE study [3], see Fig. 3 therein, the BNO expansion accelerates the convergence of the energy relative to fragment expansion. Our strategy is thus to retain the fragments at the smallest reasonable size, where most of the fragments only contain 1 atom. And the clusters are expanded with as many BNOs as practicable, up to the computational limit of our CCSD(T) calculations. ”

11. Explanations regarding the basis sets used in the calculations have been added to the captions of Fig. 2 and Fig. 3 in the main text.

For Fig. 2, the modification is “The SIE+CCSD(T) calculations are performed using cc-pV(D,T)Z extrapolated complete basis set with the neon-core correlation consistent effective core potentials [32,33]. ”

For Fig. 3, the modification is “ The SIE+CCSD calculations are performed using cc-pVDZ basis set. ”

12. The “*k*-point” in Tables S6 and S7 of the SI have been replaced with “Structure”. The words “*k*-point” in the manuscript have been removed to avoid any misunderstanding.

13. We have added the reference for the tribology applications and revised the end of Sec. 2.1 in the main text to add an explanation of quantum friction. “ the dynamics of water across graphene and subsequent implications in tribology applications [37, 44, 47]. ” “The interaction between water and a graphitic surface gives rise to many intriguing but not yet fully understood phenomena. Recent studies found that the friction for a water flow decreases as the diameter of carbon nanotubes decreases, a phenomenon known as quantum friction [37, 44, 55]. The water-graphitic interface gives rise to anomalous dielectric response [56, 57].”

14. We have restructured some sentences in the first paragraph on page 3 of the main text to improve overall clarity and remove redundancy.

“ For PBC models, the substrate is extended up to a 14×14 supercell of 392 carbon atoms, which is the closest in size to PAH(8). Both the largest OBC and PBC systems contain more than 11,000 orbitals, at which point the OBC-PBC gap is reduced to 5 meV (1 meV) for 2-leg configuration (0-leg configuration). The gap of 2-leg configuration (0-leg configuration) is further narrowed to 3 meV (below 1 meV) by considering the effects of geometry relaxation and bulk limit extrapolation in adsorption energies, given in the inset table of Fig. 2. ”

“ which explains the large OBC-PBC gaps observed in previous works, as illustrated on the left side of the Fig. 2, where high-level *ab-initio* ”

15. We have added a guidance in Sec. 2.2 of the main text directing readers to the discussion on size-consistency of SIE+CCSD(T) in SI Sec. S4.2.

“Another important reason is that SIE+CCSD(T) maintains the size consistency inherent in Coupled Cluster theory (discussed in SI Sec. S4.2), which is crucial for achieving consistent high accuracy across systems of different sizes.”

16. We added additional description to the caption of Fig. 2 in the main text to enhance clarity of the conveyed information.

“Fig. 2: **The adsorption energy of water on graphene at the bulk limit.** The computed interaction energies (ΔE_{int}) of H₂O@ Graphene are shown for both 0-leg (green) and 2-leg (purple) configurations (shown top right) as well as with open (circles) and periodic (triangles) boundary conditions. Left side displays previous quantum many-body results from Ref. [31] (superscript *a*) with the method

and the OBC–PBC gap given, with the backdrop showing the largest OBC system ($\text{H}_2\text{O}@PAH(2)$). The middle of the figure shows the SIE+CCSD(T) results with the backdrop showing our largest OBC model ($\text{H}_2\text{O}@PAH(8)$). The right side presents the final adsorption energies, ΔE_{ads} , obtained by considering both bulk limit extrapolation and geometry relaxation, with their values given in the inset table. The infinity symbol on the x-axis represents an extrapolation to the bulk limit. The details for the bulk limit extrapolation can be found in SI Sec. S3.8 and for geometry relaxation in SI Sec. S3.9. The SIE+CCSD(T) calculations are performed using cc-pV(D,T)Z extrapolated complete basis set with the neon-core correlation consistent effective core potentials [32,33].”

17. In Sec. 2.1 of the main text, we have highlighted the key role of DFT/DFTB in modeling ultra-large graphene–solvent interfaces to acknowledge the contributions of DFT-based approaches to this problem.

“Quantum many-body methods have so far been applied only to very limited graphene size [31,41]. While DFT can access much larger models [43,44] and the more efficient density functional tight binding approach can even process dynamical simulations of the water-graphitic interface [45,46], their accuracy is sensitive to the choice of underlying functionals. ”

18. We have added comments in the conclusion and outlook part to illustrate that SIE+CCSD(T) can be applied to a broader range of surface-related problems.

“For a broader range of scenarios such as cluster adsorption on surfaces, surface coverage by adsorbates at finite densities, or surface doping, SIE+CCSD(T) does not encounter any workable limitations, thereby paving the way for its application in tackling a wide array of material surface problems.”

19. Some grammatical typos and instances of improper word usage have been corrected.

20. The raw data for the interacting energies under different water orientations are added in the SI Sec. S3.11.

REPLY FOR REVIEWER 1

Comment: Huang et al. present an approach based on the systematically improvable quantum embedding (SIE) method to address problems in surface chemistry. The methodology is developed and applied at the SIE+CCSD(T) level to water on graphene surfaces to determine differences in binding energies with orientation, as well as other small carbonaceous molecules on various surfaces. The calculations performed here achieve impressive scaling with system size while also providing insight into the strength and nature of these binding interactions, in particular reaching sub-chemical accuracy in agreement with experimental data for the carbonaceous systems.

While this work introduces an interesting methodology and applies it to several chemical systems, we suggest that the work would be a more appropriate fit in another journal, such as the Journal of Chemical Theory and Computation or the Journal of Physical Chemistry C with several edits. As currently submitted, the paper may not adequately describe the novel methodological developments, and the systems evaluated may not be of appropriately broad interest. Broadly, we would suggest that this paper would be more strongly received by focusing on the methodological work and advancements in further depth in the main text, rather than the Supporting Information.

We have specifically separated our major comments for this work towards targeting each of these journals. If the authors prefer to submit the work to Journal of Chemical Theory and Computation, we suggest following comments to guide the more technically-inclined readers:

Reply: We thank the referee for acknowledging the impressive scaling of our method, the large system sizes we reached, and the sub-chemical accuracy we obtained. We are also grateful for the time invested in providing constructive, detailed, and insightful comments on our manuscript. We address all comments point-by-point below. The suggestions of the referee have unquestionably helped us improve the quality of the manuscript.

Comment: 1. Authors present a workflow that extrapolates both the open boundary condition (OBC) and periodic boundary condition (PBC) calculations to the thermodynamic limit separately to assess the finite-size errors. We ask that the authors consider explain the methodology in more detail. For example, we find the “handshake” between the OBC and PBC adsorption energies interesting and would like to see more details. We would also like to see how the differences in the bath natural orbitals (BNOs) would interact with the extrapolations. The boundary conditions likely affects the automatic BNO construction in a distinct manner, notably due to the spurious interaction in the small supercell PBC regime. We believe that inclusion of further detail will help the readers fully understand the workflow.

Reply: We appreciate the referee’s interest in the computational details of our SIE+CCSD(T) workflow, and the achievement of the OBC-PBC “handshake” in the water@graphene system.

To make the SIE workflow easier to understand, we have added further technical descriptions, and guidance on the details in the SI, as well as to the Method section of the main text (**Change 1**). To achieve the OBC–PBC handshake, some additional corrections beyond the standard SIE+CCSD(T) workflow described in the Method were still required. Although these corrections were previously included in the SI, they were somewhat scattered. Therefore, we have added Section S3.1 in the revised SI to summarize these

corrections (Change 2) to clarify the handshake. We hope these efforts will help readers better understand the technical details and appropriate context of these achievements of our work.

Regarding the influence of the BNO on the bulk limit extrapolation, we understand the referee wishes to see how changing the BNO threshold alters the extrapolated results. However, a full extrapolation would be very costly, as it would require recomputing every system size from the smallest cluster with 24 carbon atoms to the largest with 392 carbon atoms. We therefore chose two geometries with sizes already close to the bulk limit, 2-leg@PAH(6) with 216 carbon atoms for OBC and 2-leg@graphene(10) with 200 carbon atoms in PBC graphene 10 supercell. Their interaction energies are compared under the BNO thresholds from 10^{-6} to 10^{-8} to gauge the influence of this cutoff. SIE+CCSD(T) is used with the cc-pVDZ basis and ccECP, with the ratio between the water-fragment and the remaining-cluster threshold kept at $10^{1.5}$, which is commensurate with the threshold settings of the main text.

Figure R1: Interaction energy ΔE_{int} of 2-leg@PAH(6) and 2-leg@graphene(10) vs. the BNO threshold, with the inset giving the corresponding OBC–PBC gap which is defined as $\Delta E_{\text{int}}^{\text{OBC}} - \Delta E_{\text{int}}^{\text{PBC}}$.

As shown in Figure R1, for both OBC or PBC, the interaction energy rises monotonically as the BNO threshold tightens, reflecting the systematic improvability preserved by BNOs expanding in SIE workflow. The inset figure in Figure R1 shows the OBC–PBC gap. OBC–PBC gap obviously shows a much faster convergence than the interaction energy. Once the threshold drops below 10^{-7} , the change in this gap falls below 1 meV. This indicates that at sufficiently small thresholds, the interaction energy vs. system size curves used for bulk-limit extrapolation do not distort unpredictably but instead shift by a near constant offset toward the ground-truth. Throughout the SIE+CCSD(T) calculations presented in the main text, we consistently adopted a BNO threshold of 10^{-8} , and detail further energetic corrections, such as the CCSD(T)-level bath-truncation error correction. Based on the benchmark tests discussed above, we are confident that this SIE+CCSD(T) adsorption energy protocol ensures the reliability of our results. This result further underlines the careful convergence and accuracy level of our results, but does not alter the conclusions of our work. We have therefore not included the result above in the revised manuscript.

Regarding the referee’s mention of “the spurious interaction in the small supercell PBC regime”, we understand this “spurious interaction” to denote the unphysical interaction between adsorbates located in different periodic cells that arises from the use of PBC. As analysed in the main text, this interaction is

indeed the primary source of the finite-size error under PBC. Although such spurious interactions can, in principle, affect the construction of BNOs, we do not attempt to remove this influence in the BNO building step. Instead, as detailed in the manuscript, we systematically enlarge the graphene supercell in our calculations, thereby increasing the separation between the adsorbates in different periodic cells, progressively weakening their mutual interaction, and ultimately eliminating it through the subsequent bulk limit extrapolation.

Comment: 2. The authors mention that they have “implemented a GPU-accelerated version of this quantum embedding workflow...”. Technically-oriented readers will likely appreciate further detail on how the GPU-acceleration is implemented. We believe that some worded description or figures would be beneficial.

Reply: We thank the referee for their interest in our GPU implementation. All major components have been accelerated by GPU, including the multi-GPU parallel framework for SIE, the computation of cluster ERIs, and the implementations of both the high-level solver and PWF-DM. To maintain high GPU efficiency, we employed strict memory-usage control, asynchronous overlap of computation and I/O, and optimized three-tier data transfers among disk, CPU memory, and GPU memory. Further details are provided in the revised SI Sec. S2 (**Change 3**).

However, because the main focus of this manuscript is to illustrate how the SIE framework advances surface-chemistry simulations, we do not provide further code details in SI. The code is already open-source, and interested readers are encouraged to access it and run the example scripts to understand implementation specifics or directly utilize this package in their own research.

Comment: 3. The authors use “impurity” in the both the Method section (L399) and Supporting Information (eg. Page 2). Clear definition of the terminology within the context of the author’s work would be appreciated by the readers.

Reply: We thank the referee for pointing out these issues. They have now been corrected (**Change 4**). In the Method section of the main text, the term “impurity” was intended to denote the “fragment”, whereas in the SI Sec. S1.1 it referred to the “fragment+bath subspace”. All occurrences of “impurity” have therefore been replaced to avoid any misunderstanding.

Comment: 4. The authors use PWF-DM as part of the proposed workflow. We are interested in learning why the authors made a particular choice among the other schemes introduced in the paper they cite (Nusspickel et.al 2023 JCTC). We are similarly interested in a further discussion into the energy calculation, as addressed in the Supporting Information and compared to the cumulant-based methods discussed in this work.

Reply: At the beginning of SI Sec. S1.2, we briefly cite the four approaches for evaluating energy expectation values in quantum embedding which is proposed and summarized by Nusspickel et al. in [Nusspickel, Max et al., JCTC, 2023 19 2769], discuss their merits, and explain the reason why PWF-DM was adopted in this work (**Change 5**).

In brief, although Democratic partitioning of the cumulant (DPDM-C) also incorporates cross-cluster information to capture the long-range interactions between clusters, it merges clusters by directly stitching

together fragments' cumulants, just similarly to Democratic partitioning of density matrices (DPDM). The resulting global density matrix is therefore not N -representable, which means no many-electron wavefunction can reproduce it. As a result, some fundamental quantum numbers, such as the total electron number, cannot be conserved, thus introducing uncertainty into the final energies. DPDM-C must then rely on extra iterative fittings—for example, adjusting a chemical potential to enforce total electron number conservation—which adds substantial computational cost, and the optimization may become unstable when several fitting loops for conservation of several quantum numbers are required.

In contrast, the family of partitioned wavefunction methods constructs the global solution directly from the partitioned wavefunctions of the clusters, and is therefore intrinsically N -representable. No additional fitting is needed, and the quantum numbers are preserved by construction. The partition wavefunction in exponential form (PWF-EF) evaluates the correlation energy for each fragment by applying the standard configuration-interaction energy formula to the fragment-projected amplitudes, and then sums these fragment contributions to obtain the total correlation energy. PWF-DM extends PWF-EF by adding cross-cluster information. It makes PWF-DM strictly superior to PWF-EF, but at only a modest additional computational cost whose scaling does not exceed that of the SIE framework itself. For these reasons, we adopted PWF-DM in this work. This accuracy was benchmarked extensively in the work cited above. Furthermore, due to the fundamental differences in the workflow which are required for use of the DPDM-C and DPDM energy estimators (i.e. requirement of chemical potential fitting and fragments spanning the total space), these have not been implemented in our codebase, and are not expected to be given their deficiencies in performance.

Comment: 5. From what we understand, the perturbative (T) correction in SIE embedding is a novel technical contribution in this work. We are interested in seeing a benchmark for this new technique, especially as it appears that SIE+CCSD performs worse than SIE+MP2 and perturbative (T) correction has a large impact. We also ask that the authors comment on their choice not to include the cross-cluster information in constructing the (T) correction.

Reply:

We thank the referee for the positive assessment of our perturbative (T) implementation.

The referee requested a benchmark for SIE+CCSD(T), and we have provided one in SI Sec. S3.7. This benchmark was performed on the 2-leg@PAH(2) and 0-leg@PAH(2) systems using the cc-pV(D,T)Z basis sets with ccECP, extrapolated to the CBS. We compared the interacting energies obtained from SIE+CCSD(T) and canonical CCSD(T) under different BNO threshold settings. As shown in Figure R2 (which is also the Figure S5 in the SI), the results of SIE+CCSD(T) systematically approach those of canonical CCSD(T) as the BNO threshold decreases, exhibiting a clear convergence trend. This indicates that, with a properly chosen BNO threshold, SIE+CCSD(T) is expected to achieve accuracy comparable to that of canonical CCSD(T).

About the reason for the finding that SIE+MP2 can yield adsorption energies closer to the benchmark than SIE+CCSD, we believe this behaviour stems from the intrinsic properties of the high-level solvers rather than from the SIE framework itself. Although CCSD is categorically more accurate than MP2 for absolute total energies, the error-cancellation effects inherent in relative energies, such as adsorption energies, can sometimes make MP2 appear more “accurate” than CCSD. A similar outcome was reported

Figure R2: In 2-leg/0-leg@PAH(2), as the BNO threshold is reduced, the variation of interacting energy calculated by SIE+CCSD(T) (green solid line) relative to canonical CCSD(T) (red line), canonical CCSD (black line), and canonical MP2 (blue line) are shown.

by Ye et al. in the Fig. 2A of [Ye H-Z et al., Faraday Discussions, 2024 254 628], where MP2 outperformed CCSD, with CCSD(T) as the reference, for CO adsorption on MgO using the LNO approach. By contrast, in our water on graphene case, as shown in Figure R2, canonical MP2 overbinds the interaction between the water monomer and graphene while CCSD underbinds. Here the absolute error of CCSD is still slightly smaller than that of MP2 relative to CCSD(T).

With respect to introducing cross-cluster information into the perturbative (T), we have analysed its feasibility at the theoretical level. The corresponding scheme, termed the *ex-situ* form, is now outlined in the SI Sec. S1.3.2 (Change 6). Although the *ex-situ* form is formally viable, its practical implementation faces two major obstacles.

1. **Storage requirements.** Preparing the intermediates for perturbative (T) would require writing the full cluster T_2 amplitudes together with the L_2 amplitudes obtained from solving the Λ -equation in CCSD. The disk usage scales as $\mathcal{O}(mn_{\text{occ}}^2 n_{\text{vir}}^2)$, where m is the number of clusters and n_{occ} (n_{vir}) denote the numbers of occupied (virtual) orbitals in the cluster space. In the *in-situ* form, only fragment-projected amplitudes are stored, i.e., one occupied index of each T_2 and L_2 is projected onto the fragment space whose size could be considered as the constant number. The storage therefore scales as $\mathcal{O}(mn_{\text{occ}} n_{\text{vir}}^2)$, so the *ex-situ* form entails an extra order of magnitude in disk usage.
2. **Computational cost.** As shown in Eqs. S61-S63 of the SI, every pair of clusters involves an $\mathcal{O}(n^7)$ tensor contraction, with n the cluster size. Incorporating all cross-cluster terms thus scales as $\mathcal{O}(m^2 n^7)$. The *in-situ* form, by contrast, requires only the standard perturbative (T) evaluation per cluster, i.e., $\mathcal{O}(mn^7)$. As mentioned in SI Sec. S1.6, a CCSD(T) calculation on a 640-orbital cluster already takes about 60 hours on a single A100 GPU, most of which is spent in the (T) step. Considering this, an additional order of magnitude for perturbative (T) scaling would make it prohibitively expensive. It is likely that locality arguments can be used to reduce this formal scaling and only include the dominant cross-cluster contributions, but this would require further theory and code development.

For these reasons we did not adopt the ex-situ form, and believe that a more economical strategy is needed to incorporate cross-cluster information into perturbative (T). Furthermore, we physically expect higher-excitation contributions corresponding to the stronger correlations to be increasingly local, and cross-cluster contributions to the correlation energy to therefore be of less significance than the lower-rank connected contributions between clusters. However, this is not a rigorous argument and has not been numerically tested. It should be stressed that even the purely in-cluster (T) contributions will converge to the canonical (T) value as the bath space is enlarged, and therefore this is just a question of maximizing the rate of convergence. However, this is clearly an interesting route for further investigation in the future to optimize the workflow further.

Comment: 6. In Figure 2, the pipeline for the calculations implies nested, additive corrections ...

Reply: We thank the referee for raising this point. The Method section in the main text has been expanded to clarify the entire workflow (**Change 1**). We believe this modification improves the readability of our SIE workflow.

Comment: If the authors would prefer to submit the work to the Journal of Physical Chemistry C, we suggest the following comments of interest to a materials-focused audience:

1. On page 5, the authors discuss the absorption-induced dipole moment. We would suggest that the authors consider including the absorption-induced dipole moment as a function of substrate size as a figure or subplot of a figure, as we believe that this could support the main claim that a full description of the interaction-driven effects between water and graphene are all long-ranged.

Reply: We thank the referee for the suggestion. A representative figure illustrating the adsorption-induced dipole-moment change has been added, with detailed description in the figure caption and the main text (**Change 7**).

Comment: 2. On the right side of page 5, the authors claim that "graphene does not exhibit a strong preference for any specific water orientation, which is unusual for water..." Based on what the authors specify in the Supporting Information about the CBS extrapolation of relatively limited double- and triple-zeta basis set calculations, it might be nice to include references to work in which water has exhibited an orientational preference to provide context and explain how this work is different.

Reply: To obtain the conclusion that graphene is insensitive to the orientation of adsorbed water, it is necessary to calculate the interacting energy for configurations with different water orientations on large graphene substrates, as we have done in our manuscript. This is computationally very expensive, which is why previous studies, such as in the references [Brandenburg J G, et al., JPCL, 2019 10 358] and [Voloshina E, et al., PCCP, 2011 13 12041], typically considered only 2–3 plausible low-adsorption-energy configurations and assumed these to be more favorable than others. Although those studies also observed that the energies of these configurations were close to each other, the limited number of water orientation configurations considered prevented them from drawing the same conclusion as us. This discussion has been briefly summarized and incorporated into the main text (**Change 8**).

Moreover, this orientation-insensitive phenomenon of the surface and water is also unusual, especially considering water monomer has strong polarity and easily forms hydrogen bonds. [Gruber T, et al., Phys. Rev. X, 2018 8 21043], [Al-Hamdani Y S, et al., J. Chem. Phys., 2017 147] and [Thiemann F L, et al., ACS Nano, 2022 16 10775] show that a water monomer adsorbed on a hexagonal BN (h-BN) surface, or inside an h-BN nanotube, favors an orientation in which one hydrogen atom points toward a nitrogen site to form the hydrogen bond. [Li S, et al., Phys. Rev. Res., 2023 5 23018] reports that on a series of two-dimensional transition-metal dichalcogenide surfaces the ground-state configuration consistently places both hydrogen atoms of the water molecule toward the surface. These articles are now referenced and briefly discussed in the main text (**Change 8**).

Comment: 3. On page 4, the authors state that "absorption induced electron density rearrangement is considered as another descriptor for interaction range." It may help the reader to provide references and a description of why electron density rearrangement is a good descriptor for interaction range.

Reply: We thank the referee for the suggestion. The relevant reference has been cited in the manuscript (**Change 7**). The adsorption-induced electron density rearrangement is directly visualizable, making it a useful descriptor for intuitively illustrating the approximate spatial extent of the interaction between water and graphene.

Comment: We additionally offer the following minor suggestions, irrespective of which journal the authors prefer:

1. In Figure 1b, the authors present a computational time plot for their calculations. The choice of fitting the last three points suggest that some of the computational steps saturate the computational cost scaling at a large enough system size. It would be valuable for the authors to provide a breakdown of the computational workflow and identify 1) the computationally intensive step and 2) its cost scaling for a clearer picture.

Reply: These issues have been discussed in the SI Sec. S1.6. We now more clearly direct readers to this discussion from the main text Method section (**Change 9**).

Specifically, the SIE+CCSD(T) workflow involves two computationally intensive steps. The first is the full-system MP2 calculation used for the bath truncation error correction, which scales as $O(N^5)$, where N is the total system size. The second is the cost of performing CCSD(T) calculations on m clusters, which scales as $O(mn^7)$, where n is the cluster size. These two steps compete with each other at different the system sizes, and the overall computational cost as the system size increases can be divided into three stages.

- In the first stage, for small systems, the cluster CCSD(T) calculations dominate the computational time. Here, the cluster size increases with the system size. Water@graphene systems with up to around 2,000 orbitals exhibit this behaviour.
- In the second stage, for medium-sized systems, the cluster CCSD(T) cost still dominates. However, the cluster size saturates and no longer grows significantly with the system size. In this stage, the cost of a single-cluster CCSD(T) calculation can be approximated as a constant, and the overall cluster CCSD(T) cost scales as $O(Cm)$, where C is a constant. Meanwhile, the MP2 calculation remains

low-cost compared to the cluster CCSD(T) calculations. Therefore, this stage remains dominated by cluster CCSD(T) computations, and the overall scaling is approximately linear. The scaling in this stage is clearly lower than in the previous stage, which explains the observed slowdown in scaling with increasing system size in Fig. 1b.

- In the third stage, for extremely large systems, the cost of the full-system MP2 calculation rises rapidly and eventually dominates the total computational time. In our practical calculations, the system sizes have not yet reached this stage. As discussed in the SI Sec. S1.6, based on fitted scaling behaviour, we estimate that for water@graphene systems, the MP2 calculation becomes dominant when the orbital number of the full system exceeds around 15,000. At this stage, further numerical truncations can be performed for the MP2 calculations in order to reduce this cost, but were not required at this work.

Clearly, the second stage represents the high-efficiency window of the SIE+CCSD(T) method. From our fitting in the SI Sec. S1.6, this efficient stage spans from approximately 2,000 to 15,000 orbitals for water@graphene systems. We believe this window is broad enough to cover many large-scale surface chemistry problems.

Comment: 2. The authors introduce their partition strategy on page 15 of the SI. It seems like some fragments are rather small. For example, one edge fragment contains two hydrogen atoms. Does this introduce unnecessarily inexact fragment descriptions to the calculation workflow?

Reply:

Figure R3: Each differently colored box represents a distinct kind of fragment: the red box denotes the water monomer fragment, the blue box denotes the carbon fragment in graphene, and the yellow box denotes the edge hydrogen atoms fragment.

We apologize for the confusion caused to the referee. What we intended to deliver is that, in the OBC calculations, all hydrogen atoms along one of the hexagonal PAH six edges are treated collectively as a single edge fragment, as demonstrated in Figure. R3 (which is also the Figure S3 in the SI). Therefore, the size of each edge fragment increases with the size of the PAH. For a PAH(h) structure, each edge fragment contains h hydrogen atoms.

Additionally, it is true that the fragments themselves are kept relatively small. For instance, for the substrate, we assign each carbon atom to its own fragment. This design choice is discussed in detail in the

original SIE paper [Nusspickel M, et al., Phys. Rev. X, 2022 12 11046]. As explained around Fig. 3 of that work, because the bath (BNO) construction already incorporates correlated long-range information, and therefore expanding the cluster space with BNOs is a more efficient way to capture correlation than simply enlarging the fragment size, while retaining the exact limit. Therefore, our strategy is to maintain fragment sizes as small as reasonable while extending the cluster space by including as many BNOs as allowed by the CCSD(T) computational limit of our package. This also avoids having to define the choice of increasing fragment size, which would be non-unique.

Comment: 3. We request that the authors specify their choice of basis sets in the captions of Figures 2 and 3 to guide the readers.

Reply: We thank the referee for the suggestion. The basis sets used have been specified in the captions of Fig. 2 and Fig. 3. For Fig. 2, all calculations were performed with the cc-pV(D,T)Z basis sets using ccECP and extrapolated to the CBS limit. For Fig. 3, all calculations employed the cc-pVDZ basis set. It should be noted that the use of ccECP in Fig. 2 was not motivated by saving computational cost, but rather because using the conventional cc-pV x Z basis sets directly led to convergence issues in PBC CCSD calculations, which were resolved upon adopting ccECP (**Change 11**).

Comment: 4. Consider starting a new paragraph when beginning to discuss the 0-leg configuration on the left side of page 5.

Reply: We thank the referee for the suggestion. The description in this paragraph has been revised accordingly (**Change 7**).

Comment: 5. The authors outline that they used #-point calculations only on Page 12 of the Supporting Information. Is k -point sampling a direction that the authors may find interesting to explore?

Reply: We apologize for the potential confusion here. The “ k -point” specified in the SI Tables S6 and S7 refers to the size of the graphene used to construct the supercell. In fact, in Figure S2d, the PBC calculation of water@graphene used a $16 \times 16 \times 1$ graphene supercell, and all PBC calculations in this study were performed at the gamma point of the constructed supercell. Therefore, now the “ k -point” has already been removed and replaced by “Structure” in Tables S6 and S7 and all words of “ k -point” (**Change 12**) in our manuscript has been removed.

Additionally, the authors are indeed interested in exploring direct k -point sampling for computation and are considering its implementation in our package in the future.

Comment: 6. The authors discuss the BSSE errors and the counterpoise correction in Page 9. We are interested in reading if the authors made any modifications to the conventional counterpoise correction to account for the distinct automatically chosen bath natural orbitals.

Reply: We did not modify the conventional CP method, nor did we make any adjustments or changes to the methodology of BNO building. We used the canonical CCSD(T) method to verify the reliability of our workflow for calculating interacting energies using the CP method, as shown in Figure S5 of the SI.

Comment: 7. Page 4 mentions tribology applications. The reader might be interested in slight elaboration on the relevance of this application, perhaps with a citation. Paragraph 1 of page 3 provides numeric values for the OBC-PBC gaps for the 2-leg and 0-leg configurations on the left side of the page, and on the right side of the page provides differences between the OBC and PBC models for the 0-leg and 2-leg configurations. We would appreciate if the authors check for redundancy and consider restructuring this paragraph for clarity.

Reply: We thank the referee for the suggestion and have revised the relevant expressions accordingly.

We apologize for not including a citation when first mentioning tribology applications. The relevant references have now been added (**Change 13**), and we have also included an explanation of quantum friction at the end of that Section. Quantum friction refers to the intriguing phenomenon where the friction coefficient decreases as the diameter of carbon or BN nanotubes becomes smaller when water flows through them. We have cited relevant references like [Kavokine N, et al., Nature, 2022 602 84], [Thiemann F L, et al., ACS Nano, 2022 16 10775] and [Secchi E, et al., Nature, 2016 537 210]. What we intend to deliver is that phenomena such as quantum friction at the water–graphene interface or anomalous dielectric responses require a method capable of capturing the subtle electronic structure changes induced by their interactions, and our calculations demonstrate that SIE+CCSD(T) is capable of doing so (**Change 13**).

In addition, we have added and revised several sentences in the first paragraph on page 3 of the main text to enhance overall readability (**Change 14**).

Comment: 8. On the top of page 6 the authors state "SIE+CCSD(T) maintains the size consistency inherent in Coupled Cluster theory..." The reader may find systematic calculations that demonstrate size consistency useful.

Reply: We thank the referee for the suggestion. We have used organic molecules@coronene as an example to demonstrate size consistency of SIE+CCSD(T), and this analysis has been included in the SI Sec. 4.2. We have emphasized and cited this part in the main text to guide readers accordingly (**Change 15**).

As shown in Figure S13 of the SI, the deviation from experimental values increases with the molecular weight of the organic molecules, indicating that it becomes more difficult to maintain size-consistent accuracy. In contrast, SIE+CCSD(T) remains within a relatively reasonable range, demonstrating that SIE+CCSD(T) retains size consistency to a certain extent.

Comment: 9. We find Figure 2 to be a bit challenging to read: we would suggest making the figure more legible.

Reply: We appreciate the referee's suggestion, and we have incorporated additional descriptions into the caption of Fig. 2 accordingly (**Change 16**). In this figure, we aim to highlight the sizes of the systems we computed and the converged OBC-PBC gap compared to the results obtained using quantum many-body methods with SIE+CCSD(T). We believe these caption updates will enhance the clarity and understanding of the figure.

Comment: 10. We would like to point out a few minor typos: (a) (Page 8, Line 366) sysetm MP2; (b)

(Page 8, Line 371) Samething.

Reply: We appreciate the referee's careful review. All typos have been corrected in the revised manuscript (**Change 19**).

REPLY FOR REVIEWER 2

Comment: In this submission to Nat. Commun., the authors build upon state-of-the-art correlated wavefunctions to reliably converge to the ‘gold standard’ accuracy in quantum chemistry for application to extended surface chemistry. The authors use graphics processing unit acceleration along with systematically improvable multiscale resolution techniques to achieve linear computational scaling up to 392 atoms in size. The authors provide a new benchmark for this water-graphene interaction that clarifies the preference for water orientations at the graphene interface (further comments on this are given below). This is extended to the adsorption of carbonaceous molecules on chemically complex surfaces, including metal oxides and metal-organic frameworks. The authors conclude that their work enables more reliable and improvable approaches to first-principles modeling of surface problems at an unprecedented scale and accuracy using ab-initio quantum many-body methods.

I consider this work to be of interest to computational chemistry/materials researchers as well as readers of this journal. As such, I am generally supportive of publication with a few minor edits.

Reply: We sincerely thank the referee for the very positive assessment of our work and supportive of publication with proper revision. We appreciate the recognition of our efforts to achieve the systematically improvable and reliable modeling of extended surface chemistry at the “gold standard” CCSD(T) level. We are also grateful for the referee’s thoughtful reading and valuable comment, which has helped us further strengthen the presentation of our manuscript. We believe that with these improvements, the work can better contribute to the broader computational chemistry and material surface communities.

Comment:

The authors clearly have a capability that is superior to other "empirical" approaches such as DFT. However, there has been previous work in the field using DFTB to treat large systems, which should be noted: J. Phys. Chem. C 2016, 120, 19212–19224 and J. Am. Chem. Soc. 2024, 146, 35313–35320. Specifically, these prior studies also had the same goal of linear scaling of large solvated systems (although the treatments in these studies are not at the same level of accuracy of the work under study). In conclusion, I consider this work to be impressive, but the authors should also mention some of the historical development in this field of linear scaling for large solvated systems.

Reply: We have appropriately referenced prior studies in the field (**Change 17**), as suggested by the referee. Specifically, we are impressed by the effectiveness of DFTB in modeling graphene-solvent interfaces, as exemplified by references [J. Phys. Chem. C 2016, 120, 19212–19224] and [J. Am. Chem. Soc. 2024, 146, 35313–35320]. These references are cited in our manuscript to highlight DFTB advancements for large solvated systems.

REPLY FOR REVIEWER 3

Comment: In this manuscript, the authors extended the "systematically improvable quantum embedding" (SIE) approach to achieve CCSD(T) level accuracy for the binding energy between molecules and substrates. Given the importance of energetics in surface science (such as catalysis), this work will open new doors for accurate determinations of binding geometries and binding energies and will be of broad interest. The paper is nicely written, with sufficient technical details in the Supporting Information. I recommend publication in Nature Communications, with the following comments for the authors to consider.

Reply: We sincerely thank the referee for their positive evaluation of our work and for recommending its publication in Nature Communications. We are grateful for their recognition of the potential impact of our extension of the systematically improvable quantum embedding (SIE) approach to achieve CCSD(T) accuracy in surface science applications. We also deeply appreciate the thoughtful comments and suggestions provided. These insights have been invaluable in helping us further improve the quality and clarity of the manuscript. In the following, we provide detailed responses to each of the referee's comments and describe the corresponding revisions made to the manuscript.

Comment: (1) On page 9 of the main text, at the beginning, the authors mentioned that the geometry relaxation effect is typically small and can be captured at the DFT level. I am not sure I fully understand this. There are two related questions: (i) Can the authors justify this statement based on some explicit calculations, even for a very small system (I understand that the geometry optimization is expensive with SIE, as the authors mentioned in Sec. 2.8 of the SI). (ii) When one uses DFT to capture this effect, what is the recommended procedure or functional? In Sec. 2.8 of the SI, the authors mentioned that B97M-V and ω B97M-V are used for the systems studied in this work: does this conclusion apply generally to other systems, or is there a recommended procedure for finding the "optimal" functional?

Reply: We appreciate the referee's inquiries. We understand that the referee is concerned that DFT may not be capable of producing accurate relaxed geometries and hopes that we can verify whether certain DFT functionals can truly achieve accuracy comparable to high-level methods such as CCSD(T). As mentioned in the revised SI Sec. 3.9, reaching CCSD(T)-level accuracy is expensive even using SIE, and for this reason, we did not implement geometry optimization with SIE in our code. While the referee suggests verification on small systems, we believe this approach is also not feasible. Even if DFT yields results consistent with CCSD(T) on small systems, it does not guarantee that DFT can correctly describe larger systems. For example, in the case of water@graphene, we have demonstrated that long-range van der Waals interaction plays a crucial role in adsorption. However, this interaction is absent in small models, such as water@benzene, due to the limited substrate size, making it unreliable to judge a functional's applicability to large systems if the benchmark is carried on such small model.

A more appropriate strategy is to select a suitable functional based on comparison with high-level calculations of the property of interest on a moderate-size system. As described in the SI Sec. S3.9, we use the interacting energy calculated by SIE+CCSD(T) for the water@PAH(4) system as the reference data, identify suitable functionals, and subsequently find B97M-V and ω B97M-V can yield similar results. This protocol is, to some extent, recognized by the community and has been widely adopted, as evidenced by

similar methodologies used in [Shi B X, et al., J. Am. Chem. Soc., November 2023 145 25372], [Ajala A O, et al., J. Chem. Theory Comput., 2019 15 2359] and [Brandenburg J G, et al., J. Phys. Chem. Lett., 2019 10 358].

Several commonly used softwares that support DFT calculations allow for the selection of functionals and evaluation of geometry relaxation effects, such as VASP, Quantum Espresso, PySCF, or GPU4PySCF. The general functional selection process typically follows the protocol below.

- Firstly, the protocol starts by using a structure that can be computed accurately with a high-level method and using the results from this high-level method as a reference. Ideally, this high-level method is a state-of-the-art method recognized within the community, such as CCSD(T) or a method with equivalent accuracy (this study directly employs SIE+CCSD(T)), or QMC, RPA, etc.
- Secondly, a set of functionals is chosen to benchmark against the reference. The selection of a set for functionals should consider the nature of the system to increase the likelihood of obtaining appropriate functionals. As discussed in our paper, for systems like water@graphene, the long-range interaction between water and graphene is crucial. Functionals like B97M-V and ω B97M-V were chosen because they were designed to capture such long-range interactions. Other functionals, such as B3LYP or the PBE series, are well-known functionals that, with the addition of long-range corrections like D3, also have the potential to effectively describe systems like water@graphene, and therefore were included.
- Finally, the most suitable functional is the one that minimizes the deviation from the reference.

It is important to note that constructing a test set of functionals requires prior knowledge, suggesting that there may not be a completely automated optimal functional selection process. For similar reasons, while B97M-V and ω B97M-V may be suitable for describing water@graphene, they may not achieve the same level of accuracy in other systems and require testing to determine their applicability. This uncertain and non-universal choice underlines the importance of systematically improvable approaches as exemplified in this work.

Comment: (2) It appears to me that in all the systems studied in this work, there is a single molecule adsorbed on a large-area substrate surface, and one needs to check the convergence to the "bulk" limit (where the surface is infinitely large). However, a more common experimental scenario is that, there is a finite coverage, i.e., one adsorbed molecule per $X \text{ nm}^2$ (X is a number that depends on the experimental condition). The coverage can be very low in some cases, but the "bulk" limit discussed in this work does not seem to be physical. The authors should comment on this point, and should also comment on whether this approach can be extended to cases to accommodate the finite coverage (presumably, treating the molecular adsorbates as periodic images).

Reply: We appreciate the referee's comment. Simulations of both a single adsorbate on an infinitely large substrate, as well as simulations with adsorbates covering the substrate at a certain density, are important but serve different purposes.

In the former, monomer adsorption simulation is critical for surface catalysis, such as single-atom catalysis. For example, the adsorption energy of a monomer on a surface is essential for determining the

likely site of an interfacial reaction. Additionally, the interaction range between the monomer and the surface can be used to infer suitable reactant densities for effective reaction occurrence.

In the finite coverage case, as suggested by the referee, modeling and simulating adsorbates covering the surface at a specific density within a PBC cell are appropriate, such as the heterogeneous catalysis process. In both cases, SIE+CCSD(T) to simulate these models remains viable by directly running SIE+CCSD(T) using PBC model. This statement has been added to the conclusion part in the main text (**Change 18**).

Comment: (3) Throughout the manuscript, there are a few places where "adsorption" was spelled incorrectly as "absorption" and should be corrected.

Reply: We thank the referee for this careful reading and for pointing out these typos. We have modified them accordingly in the revised manuscript (**Change 19**).